# Phytochemical Analysis, α-Glucosidase and α-Amylase Inhibitory Activities and Acute Toxicity Studies of Extracts from Pomegranate *(Punica granatum)* Bark, a Valuable Agro-Industrial By-Product

**DOI:** 10.3390/foods11091353

**Published:** 2022-05-06

**Authors:** Nassima Laaraj, Mohamed Bouhrim, Loubna Kharchoufa, Salima Tiji, Hasnae Bendaha, Mohamed Addi, Samantha Drouet, Christophe Hano, Jose Manuel Lorenzo, Mohamed Bnouham, Mostafa Mimouni

**Affiliations:** 1Laboratory of Applied Chemistry and Environment (LCAE), Faculty of Sciences Oujda (FSO), University Mohammed First (UMP), Oujda 60000, Morocco; nassima.laaraj1@gmail.com (N.L.); salimatiji@gmail.com (S.T.); hasnae.bendaha@gmail.com (H.B.); 2Laboratory of Bioresources, Biotechnology, Ethnopharmacology and Health, Faculty of Sciences Oujda (FSO), University Mohammed First (UMP), Oujda 60000, Morocco; mohamed.bouhrim@gmail.com (M.B.); kharchoufa.loubna@ump.ac.ma (L.K.); mbnou-ham@yahoo.fr (M.B.); 3Laboratoire d’Amélioration des Productions Agricoles, Biotechnologie et Environnement, (LAPABE), Faculté des Sciences, Université Mohammed Premier, Oujda 60000, Morocco; m.addi@ump.ac.ma; 4Laboratoire de Biologie des Ligneux et des Grandes Cultures, INRAE USC1328, Orleans University, CEDEX 2, 45067 Orléans, France; samantha.drouet@univ-orleans.fr; 5Centro Tecnológico de la Carne de Galicia, Rúa Galicia Nº 4, Parque Tecnológico de Galicia, San Cibraodas Viñas, 32900 Ourense, Spain; jmlorenzo@ceteca.net; 6Área de Tecnología de los Alimentos, Facultad de Ciencias de Ourense, Universidad de Vigo, 32004 Ourense, Spain

**Keywords:** *Punica granatum* bark, phytochemical investigation, sequential extraction, antidiabetic activity, α-glucosidase and α-amylase enzymes, acute toxicity

## Abstract

*Punica granatum* is a tree of the Punicaceae family which is widespread all over the world with several types of varieties. Its fruit juice is highly prized, whereas the bark, rich in in phytochemicals such as flavonoids, hydrolysable tannins, phenolic acids, and fatty acids, is regarded an agro-industrial waste. It is utilized in traditional medicine for its medicinal properties in the treatment and prevention of a variety of ailments. This study aims to extract and to separate the phytochemical compounds from the bark of *P. granatum*, to identify them and to study the inhibitory effect of its extracts against antidiabetic activity. First, we carried out successive hot extractions with solvents (chloroform, acetone, methanol, and water) of increasing polarity by the Soxhlet. Then, using both qualitative and quantitative phytochemical investigation, we were able to identify groups of chemicals that were present in all extracts. We identified the majority of the molecular structures of chemicals found in each extract using HPLC-DAD analysis. The inhibition against both intestinal α-glucosidase and pancreatic α-amylase enzymes by *P. granatum* extracts was used to evaluate their potential antidiabetic effect in vitro. Our results demonstrated the great potential of the acetone extract. Ellagic acid, (−)-catechin, vanillin and vanillic acid were proposed as the most active compounds by the correlation analysis, and their actions were confirmed through the calculation of their IC_50_ and the determination of their inhibition mechanisms by molecular modelling. To summarize, these results showed that *P. granatum* bark, a natural agro-industrial by-product, may constitute a promising option for antidiabetic therapeutic therapy.

## 1. Introduction

Diabetes mellitus (DM) refers to a group of chronic diseases characterized by high blood glucose that can be called hyperglycemia [1].There are two well-known types of diabetes: T1DM represents 5% to 10% of all diabetic cases and it is caused by a deficiency of insulin production.T2DM that has a direct correlation with insulin resistance and relative insulin deficiency, is also considered as the most common type in diabetes which represents 90% to 95% of worldwide cases [2]. Currently, the prevalence of diabetes mellitus has increased from 108 million in 1980 to 425 million in 2017 and this number continues to increase; according to forethought statistics, it will increase to 629 million in 2045 [3].

Therefore, some drug therapies with antidiabetic effects such as insulin and synthetic medication are used to treat type 2 diabetes. Nonetheless, these therapeutic supports have various side effects such as hypoglycemia and weight gain which can exert a negative influence on the patient [4]. One of the treatments for decreasing postprandial hyperglycemia is related to the inhibition of digestive enzymes and the reduction in glucose intestinal absorption [5]. Intestinal α-glucosidase and pancreatic α-amylase are important enzymes for carbohydrate digestion and absorption in humans. α-amylase catalyzes the hydrolysis of the dietary starch into maltose; these disaccharides are digested further by α-glucosidase which is a specific membrane-bound enzyme in the small intestine [6]. For that reason, natural α-glucosidase and α-amylase inhibitors, from food products, have appeared as promising therapeutic options to supplement or even substitute existing drugs. Several types of natural plant products have appeared as promising α-glucosidase and α-amylase inhibitors in recent decades [7]. Recently, extracts of two natural matrices studied in our laboratory showed an underlined inhibition against these two enzymes [7,8].

Pomegranate (*Punica granatum*), which inherits its name from the Latin “*pomus*” and “granum”, which literally translates to apple with grain, is a tree belonging to the Punicaceae family [9]. This fruit is native to semitropical Asia extending throughout the Mediterranean and is now planted in India, Afghanistan, Russia, Japan, as well as in the United States, including California and Arizona [10]. All of the edible and nonedible parts of the pomegranate plant have a lot of therapeutic and pharmacological effects which are due to the presence of several phytochemical compounds such as flavonoids, hydrolysable tannins, alkaloids, phenolic acids, fatty acids and lignans [11]. The *P. granatum* bark which is considered as an agro-industrial waste, has been shown to possess healthy properties such as anti-inflammatory, anti-cardiovascular, antibacterial, anti-tyrosinase, antihyperglycemic, antihyperlipidemic, and antioxidant activity [12]. For these multiple therapeutic effects, the bark of *P. granatum* has been used extensively in traditional medicine since antiquity [13].

The aim of this work was to extract and separate the bioactive compounds from the bark of *P. granatum* using Soxhlet at a hot temperature (40 °C). For this, successive extractions were carried out in different solvents (hexane, chloroform, acetone, methanol, and water). The different polarities of these solvents allowed different types of chemical groups to be extracted and separated simultaneously. In order to specify the nature of these chemical groups and their quantities, a qualitative and quantitative phytochemical study was carried out. The compositions of the extracts were then characterized by HPLC-DAD. Afterward, the potential antidiabetic activity of pomegranate bark extracts was evaluated by inhibiting two enzymes, intestinal α-glucosidase and pancreatic α-amylase. Correlation analysis allowed us to propose some phytochemicals as potential inhibitor candidates, which were then validated by calculating the IC_50_ against intestinal α-glucosidase and pancreatic α-amylase, as well as utilizing molecular docking. Finally, the acute toxicity of the extracts was evaluated in vivo on Swiss albino mice.

## 2. Materials and Methods

### 2.1. Chemicals

All solvents (hexane, chloroform, acetone, methanol, dimethyl sulfoxide (DMSO)), intestinal α-glucosidase, pancreatic α-amylase, and other reagents were purchased from Sigma-Aldrich (Waltham, MA, USA). All products used were of analytical grade.

### 2.2. Plant Material

Fresh pomegranate fruits were harvested in October 2021, and obtained from a local market (Oujda, Morocco). The bark was manually separated, cut into small portions, dried in an oven at 40° for 48 h, and ground to obtain a fine powder using a Blender Moulinex Type 320.2.00. This temperature and drying time were necessary and sufficient to preserve and keep the chemical constituents of pomegranate peel powder unchanged during all research work. As a result, the fine powder was stored away from light at 4 °C in borosilated glass bottles for future use.

### 2.3. Preparation of the Punica granatum Extract

About 50 g of pomegranate bark powder was prepared for extraction using five solvents of different polarities (hexane, chloroform, acetone, methanol, and water) in a Soxhlet apparatus, then extracts were recovered using a rotary evaporator and conserved in a refrigerator at 4 °C for further studies (Figure 1). The yields extract was calculated according to the mass of pomegranate bark put in the Soxhlet apparatus.

### 2.4. Phytochemical Investigation of Punica granatum Bark (PGB) Extracts

Qualitative and quantitative phytochemical analysis was carried out according to known standard protocols and tests in order to specify all chemical groups and compounds contained in the various extracts obtained.

#### 2.4.1. Preliminary Phytochemical Screening

Steroids and triterpenoids trials: Each dried extract was dissolved in chloroform and then acetic anhydride. Two or three drops of concentration H_2_SO_4_ were added into the mixture. During the reaction, the color changed from red to blue and to dark green which indicated the presence of steroids and triterpenoids, respectively [14]. The crude extract was dissolved in 1 mL HCl (50% *v*/*v*). After that, Mayer reagent was added and the presence of alkaloids was marked by the formation of a yellow precipitate [15]. The Shinoda test was used by adding a few magnesium turning and concentrated hydrochloric acid in PPE, and the color change from red to pink confirmed the presence of flavonoids [16]. The presence of polyphenol and tannin was assessed as follows: 1 mL of the extract solution was treated with a volume of 1 mL of 10% ferric chloride and the appearance of a green or blue color indicated the presence of polyphenols. Thus, the progressive change of color in the solution towards dark blue suggested the presence of hydrolysable tannins while the progression towards the green color showed the existence of condensed tannins [17]. For saponins, in a test tube, we put 1 mL of the extract with 6 mL of distilled water which was agitated vigorously with a vortex for a few minutes; the formation of a persistent foam confirmed the presence of saponins [18]. The presence of coumarin was evidenced in a test tube. A total of 2 mL of extract was mixed with 3 mL of basic solution (sodium hydroxide, 10% NaOH); the formation of a yellow coloration proved the presence of coumarins [19]. For anthraquinones: a few drops of an ammonia solution (10%) were added to 1 mL of the extract; the appearance of a pink color or of a precipitate confirmed the presence of anthraquinones [20].

#### 2.4.2. Quantitative Phytochemical Analysis

Evaluation of phenolic compounds present in the extracts was carried out by assay with the Folin–Ciocalteau reagent [21]. In a test tube, 200 μL of the extract was mixed with 1 mL of the Folin–Ciocalteau reagent which was incubated for 4 min, after which 800 μL of Na_2_CO_3_ solution (7.5 g/L) was added. The mixture was kept in the dark at room temperature for 1 h, then the absorbance was measured at 700 nm using a UV-VIS spectrophotometer. Gallic acid standard solutions were prepared previously and the calibration curve was plotted. Total phenolic contents were expressed as the gallic acid equivalent (GAE, milligrams per 100 g of dry weight (DW)).

The determination method of the flavonoid content was performed according to the procedure described in the literature by Dalli et al. [22]. To 200 μL of the sample, 1 mL of distilled water and 50 μL of sodium nitrate (5% *w/v*) were added. After 5 min of evolution at room temperature, 120 μL of AlCl_3_ (10% *w/v*) was added and the samples were incubated again for 5 min. Afterwards, 400 μL of NaOH (1 M) was added and then their absorbances were measured at 430 nm. Quercetin was used as a standard to construct the calibration curve; for this, several standard solutions were prepared beforehand and their absorbances were measured by UV-Visible. The flavonoid content was expressed as the quercetin equivalent (QE, mg per 100 g of dry weight (DW)).

### 2.5. GC-MS Analysis

The chemical composition of the Hexane bark extract of *P. granatum* was analyzed using gas chromatography coupled with Mass Spectroscopy ((GC-MS-QP2010, Shimadzu, Japan) equipped with a BPX25 capillary column (30 m × 0:25 mm, 0.25 μm film thickness). Helium gas was used with a constant flow rate of 3 mL/min, then the interface and ion source temperatures were maintained at 250 °C. Electron ionization was performed at 70 eV: about 1 μL of the sample was injected in a spitless mode. The identification and quantification of the compounds was determined by comparing retention indices and spectral mass fragments from a computer library. Fatty acid components were identified by comparing retention times and mass spectral fragments with standards using National Institute of Standards and Technology (NIST147).

### 2.6. HPLC-DAD Analysis

The phenolic compounds present in the chloroform, acetone, methanol, and water extracts of the *Punica granatum* bark were identified using the Waters Alliance 2695 HPLC separation module with a photodiode array detector (2998). A C18 column (5μm, 250 × 4.6 mm) was used. The elution was performed by a binary gradient system (A: water/acetic acid (2% *v/v*) and B: Methanol was prepared as the mobile phase with a flow rate of 1 ml/min). The gradient method was applied as reported by [23]: 5% B for 5 min, 5–70% B for 25 min, and 70–5%B for 10 min. The samples were filtered through a 0.45 μm filter and the injection volume was adjusted at 20 μL. By default, the detection was set at 254 nm. The retention time and UV spectra of the extract matched with our internal standards library: gallic, caffeic, *p*-coumaric, vanillic, and ellagic acids, as well as rutin, catechin, vanillin, punicalagin, and phlorizin.

### 2.7. Antidiabetic Activity of the Fruit Bark Extract of Punica granatum

#### 2.7.1. In Vitro Inhibition of Intestinal α-Glucosidase

The inhibitory activity of the pomegranate bark extract against intestinal α-glucosidase was evaluated following the release of glucose from sucrose and the experimental method was explained by Ouassou et al. [24]. α-glucosidase, sucrose, and glucose solutions were dissolved in a phosphate buffer at pH 7.5 to obtain α-glucosidase (10 UI/mL), sucrose (50 mM), and glucose (1 g/L), respectively. In addition, all extracts dissolved in DMSO were tested at two concentrations (166 and 332 μg/mL).

To realize this assay, 10 μL (166 and 332 μg/mL) of the extracts or positive control (Acarbose) or negative control (distilled water) were mixed with 0.1 mL of α-glucosidase solution (10 IU), 1 mL of phosphate buffer (50 mM; pH =7.5), and 0.1 mL of sucrose (50 mM). The mixtures were incubated at 37 °C for 25 min. The reaction was stopped by heating test tubes for 5 min at 100 °C in a water bath. The release of glucose was determined by the glucose oxidase (God-Pod) method using a commercially available kit. Lastly, the absorbance was read at 500 nm.

The percentage of inhibition was computed using the following formula:(1)$$\mathrm{Inhibitory}\;\mathrm{activity}\;\%=\frac{\mathrm{Abs}\;(\mathrm{control}\;500\;\mathrm{nm})-\;\mathrm{Abs}\;(\;\mathrm{sample}\;500\;\mathrm{nm})}{\mathrm{Abs}\;(\mathrm{control}\;500\;\mathrm{nm})}\times 100$$

Abs_Control 500 nm_: Absorbance of the control without inhibitor.

Abs_Sample 500 nm_: Absorbance of sample contained *P. granatum* extracts or Acarbose.

#### 2.7.2. In Vitro Inhibition of Pancreatic α-Amylase

Pancreatic α-amylase inhibition activity was assessed according to the method described by Daoudiet al. [25]. A total of 200 μL of PPE at two concentrations (0.5 and 1 mg/mL) or Acarbose (positive control) or phosphate buffer (control) was homogenized with 200 μL of α-amylase enzyme. The solutions were prepared by dissolving the enzymes and samples in a phosphate buffer solution (0.1 M, pH 6.9). The mixtures were incubated for 10 min at 37 °C. Next, 200 μL of 1% starch also dissolved in a phosphate buffer was added to the reaction mixture, and all test tubes were incubated for another 20 min at 37 °C. To stop the reaction, 600 μL of DNSA (1 g of 3,5-dinitrosalicylic acid dissolved in 20 mL of NaOH(2N), combined with 30 g of KNaC_4_H_4_O_6_, and filled up to 100 mL with distilled water) was added. Afterward, the mixture was posed in a water bath at 100 °C for 8 min, then it was put in an ice-cold-water bath for a few minutes. About 1 mL of distilled water was added to the sample tubes before their absorbance measurement at 540 nm.

The percentage of inhibition was computed using the following formula:(2)$$\mathrm{Inhibitory}\;\mathrm{activity}\;(\%)=\frac{\mathrm{Abs}\;\left(\mathrm{control}\;540\;\mathrm{nm}\right)-\;\mathrm{Abs}\;\left(\;\mathrm{sample}\;540\;\mathrm{nm}\right)}{\mathrm{Abs}\;\left(\mathrm{control}\;540\;\mathrm{nm}\right)}\times 100$$

Abs_Control 540 nm_: Absorbance of the control without inhibitor.

Abs_Sample 540 nm_: Absorbance of sample contained *P. granatum* extracts or Acarbose.

#### 2.7.3. IC_50_ Determination

Increasing concentrations were utilized to calculate the IC50 value. The enzymatic assays were carried out as indicated above. The Quest Graph™ ED50 Calculator (AAT Bioquest Inc., Sunnyvale, CA, USA, https://www.aatbio.com/tools/ed50-calculator, accessed on 1 February 2022) was employed to calculate the IC_50_.

#### 2.7.4. Molecular Docking Analysis

The molecular docking analysis was performed as previously described in Tiji et al. [7]. The PyRxvirtual screening tool software was used to predict the conformation of the molecule ligands within the appropriate target binding site of pancreatic α-amylase (PDB: 2QMK) and α-glucosidase (PDB: 2QMK) using Autodock 4 and Autodock Vina (The Scripps Research Institute, La Jolla, CA, USA) and Pymol v2.1.1 (Schrodinger, New York, NY, USA) (PDB: 5NN5). Discovery Studio 2020 (Dassault Systemes, Vélizy-Villacoublay, France) was used to determine the kind of contact and to visualize it in two dimensions, while UCSF Chimera 1.14 (San Francisco, CA, USA) was utilized to depict the molecules and interaction residues in three dimensions.

### 2.8. Acute Toxicity Evaluation

#### 2.8.1. Experimental Animals

Swiss albino mice were procured from the animal laboratory of the biology department of Mohammed I University (Faculty of Sciences-Oujda, Morocco). The animals were housed in polycarbonate cages and kept under standard conditions (22–26 °C with 12 h light–dark cycle) and they were also given free access to food and water. All mice were cared for in compliance with the internationally accepted guide for the care and use of laboratory animals published by the U.S. National Institute of Health [26].

#### 2.8.2. Oral Acute Toxicity in Mice

The acute toxicity study of the pomegranate bark extract was carried out in albino mice following the instructions in the OECD guidelines [27]. Briefly, thirty-six mice (20–30 g) were divided into 6 groups with 3 males and 3 females of 6 mice each. The control group received distilled water only, whereas the experimental groups were treated with hexane, chloroform, acetone, methanol, and water extracts administered orally at a single dose of 2000 mg/kg body weight. Visual observations including weight loss, mobility, respiration, skin changes, or other behavior were made and recorded for the first 30 min and then at 1, 2, 3, 4, 5, and 6 h after administration of the pomegranate bark extract. Thereafter, toxic symptom observations were noted daily until the 14th day.

### 2.9. Statistical Analysis

Data are presented as the mean ± standard errors and were subjected to statistical analysis using Graph Pad Prism 5 Software (San Diego, CA, USA). Multiple-group comparisons were analyzed by one-way analysis of variance (ANOVA). Statistical significance was accepted as *p* ≤ 0.05. Correlation analysis was performed using PAST3.0 software.

## 3. Results and Discussion

### 3.1. Extraction Yields

The extract was first defatted using hexane (HEE), and then extracted using solvents with increasing polarities. The percent yields of different extracts obtained from pomegranate bark using Soxhlet apparatus were classified in decreasing order as: water (WAE) > methanol (MEE) > acetone (ACE) > chloroform (CFC). Extraction with water gave the highest value (31%) while chloroform represented a lower yield (2%). Sequential extraction of *P. granatum* gave five extracts which were different in appearance. HEE and CFC extracts were yellow and brown solids, respectively, while the polar extracts were viscous and range color were a red orange color for ACE and MEE the extracts, and dark brown color for WAE. Our results are in agreement with those found in the literature [28,29]. On the other hand, it has been found that a methanol extract of *P. granatum* from India gave the maximum value followed by water [30,31].

### 3.2. Qualitative Screening

A preliminary phytochemical study is a helpful step in the determination of the secondary metabolite present in the *P. granatum* bark extract. The results are presented in Table 1.

Flavonoids, polyphenols, and tannins were detected in the acetone, methanol, and aqueous extracts, while steroids and terpenoids were remarkably present in the chloroform extract. It was also noted that the water and chloroform extracts of the pomegranate bark contained alkaloids and coumarins, respectively. On the other hand, all extracts showed the absence of saponins and anthraquinones.

This analysis showed the effect of different solvents on the extraction of the bioactive compounds. Non-polar extracts were found to be rich in steroids and terpenoids but polar extracts contained flavonoids and polyphenolic compounds. The studies of Sakthivel et al. [32] and Karthikeyan [19] confirmed that the Indian extract of *P. granatum* is rich in both flavonoids and polyphenolic compounds.

### 3.3. Quantitative Screening

The total phenolic compound (TPC) and total flavonoid compound (TFC) contents in the five extracts of pomegranate bark are illustrated in Table 2. The highest content of (TPC) (232.45 ± 9.67 mg GAE (gallic acid equivalent)/100 g DW)) was obtained from the methanol extract followed by the acetone and water extracts, respectively (183.92 ± 3.21), (131.55 ± 1.19) mg GAE/100 g DW. Concerning (TFC), acetone and water extract contained almost the same value (117.08 ± 5.32 mg of QE (quercetin equivalent) /100 g DW), while the methanol extract showed the highest level. On the contrary, the hexane and chloroform extracts registered the absence of these compounds; this may be linked to their lipophilic and lipophilic–hydrophilic character.

Our results are in a good agreement with previous publications [17,31,33,34], showing that MEE gave the maximum content in total polyphenols and flavonoids compared to other extracts. Variations in TPC and TFC can be explained by the types of pomegranate bark used or the different solvent systems and extraction procedures applied in the experiments.

### 3.4. HPLC-DAD Phytochemical Characterization

First, GC-MS confirmed the fatty acid composition of the defatted hexane fraction (HEE) of the pomegranate bark (Appendix A; Appendix A). These results are in good agreement with previous reports [35,36,37] and show the effective elimination of the majority of the fatty acid fraction using hexane.

The chemical composition for each extract obtained with different solvents (water, methanol, acetone, and chloroform) was determined by HPLC-DAD. The identification was carried out by comparison with the standard molecules using our internal database (Figure 2 and Figure 3); consequently, the majority of the peaks were assigned. It clearly appears that the shape and intensity of the chromatogram acetone signals were much closer to those of methanol than those of water. This is endorsed by the close amounts of (TPC) and (TFC) found on the quantitative phytochemical analysis (Table 2).

Data analysis showed the presence of various compounds which distributed into different extracts: flavonoids (rutin, naringenin, catechin,) phenolic acids (ellagic acid, gallic acid caffeic acid, *p*-coumaric acid, vanillic acid), and ellagitannin (punicalin and punicalagin isomers). It should be noted that several compounds had similar molecular structures. Others may have resulted from a sample degradation reaction (Figure 4).

The existence of ellagic, gallic acid, rutin and punicalagin in the extracts has been widely reported in the literature [30,34,38] with some differences in percentage and retention time (RT). Moreover, by HPLC analysis, several authors [39,40,41] revealed the presence of punicalagin as being the majority compound. Another result obtained by Fawole et al. [42] showed that ellagic acid was the most abundant phenolic compound detected in all seven pomegranate cultivars.

### 3.5. In Vitro Inhibition of α-Glucosidase and α-Amylase by P. granatum Bark Extracts

The inhibition measurements of both enzymes were carried out for two different concentrations (166 μg/mL, 332 ug/mL) (Figure 4). The dose–response curves are presented in Appendix A.

Foremost, it should be pointed out that the low concentration (166 ug/mL) was more efficient than the larger one (332 μg/mL). The ACE was very effective at two concentrations and had a remarkable inhibition on intestinal α-glucosidase as compared to the control with an IC_50_ value of 96.19 μg/mL (Appendix A). The percentage inhibition values of the other extracts were lower than that obtained with ACE, with IC_50_ values of 449.12, 600.25 and 662.44 μg/mL for WAE, CHE, and MEE, respectively (Appendix A). Based on this result, ACE is considered as the best inhibitor of α-glucosidase with the maximum value of 95,85% at 166 μg/mL.

In the case of α-glucosidase, ACE inhibition may be due to the presence of three molecules, punicalin, punicalaginsα, and ellagic acid. If we compare this inhibition with MEE, this is due rather to the mutual presence of punicalin (14.14%) and punicalagin (30.08%) because the percentages of the two other compounds are very similar. If we compare it to the CHE without considering the polarity difference in the solvents, the inhibition can be attributed essentially to the presence of punicalins. Therefore, this last molecule plays an essential role in the inhibition.

The inhibition potential of the extracts on the pancreatic α-amylase were evaluated in vitro at two doses (500 and 1000 μg/mL) (Figure 5). The dose–response curves are presented in Appendix A.

Our results showed that the inhibitor will increase with increasing concentrations. It is clear that the more polar extracts (ACE, WAE, and MEE) had the highest inhibitory potential with IC_50_ values of 373.90, 387.36, and 399.45 μg/mL (Appendix A) as compared to the less polar extract CHE which showed the weakest activity (IC_50_ of 436.63 μg/mL, Appendix A).

Multiple evidence has shown the undesirable side effects of Acarbose, which led to the approved choice of natural inhibitors as an interesting therapy [43]. Various studies have been published on the antidiabetic activity of the pomegranate fruit [44,45,46]. Using high-performance liquid chromatography, extracts of *P. granatum* revealed the existence of several polyphenolic compounds that can inhibit the digestive enzymes. As an example, phenolic acids like gallic acid have been reported to inhibit both *α*-glucosidase and *α*-amylase [47,48] and had an important role to reduce the aftereffects of acarbose when used in combination [49]. It was reported in Italy that the antidiabetic activity of *P. granatum* has been attributed to punicalagin [50]. Moreover, ellagic and gallic acids found in methanolic extract leaves have an antidiabetic activity [51]. Similarly, it has been confirmed that both acids had selective inhibition on intestinal *α*-glucosidase, but they exhibited weak or no effects on pancreatic *α*-amylase [52]. Among the identified compounds, rutin was found to be able to inhibit the tested enzymes, notably *α*-glucosidase [53].

The principal phytochemicals from *P. granatum* extracts were tentatively connected to the inhibition of α-glucosidase and α-amylase using correlation analysis (Figure 6).

In this study, in addition to TPC and TFC, which were shown to be significantly correlated with α-amylase inhibition, vanillic acid and vanillin were found to be significantly correlated with α-glucosidase inhibition, and catechin and ellagic acid with α-amylase inhibition.

We next determined the inhibition capacity of these compounds to validate this correlation analysis by determining the IC_50_ values of the pure commercial standards of these phytochemicals against these two enzymes. The results are presented in Figure 7 and they confirmed the inhibition potential of vanillic acid and vanillin against α-glucosidase, and of catechin and ellagic acid against α-amylase.

The IC_50_ values obtained were in the range of those previously recorded for flavonoids and/or phenolics against these two enzymes [7,8].

These results were also confirmed by molecular docking with the determination of the interaction modes. Figure 8 depicts the docking results in terms of potential binding to the human pancreatic α-amylase and the human intestinal α-glucosidase of these phytochemicals.

Both compounds were docked in the active site of human pancreatic α-amylase, indicating the possibility of competitive inhibition. The predicted affinities for (−)-catechin and ellagic acid were −9.0 kcal/mol and −8.2 kcal/mol, respectively. The small advantage for catechin may be due to more predicted hydrogen bonds. Similarly, vanillic acid and vanillin interactions were predicted to be in the active site of human intestinal α-glucosidase with very similar calculated affinities of −5.7 kcal/mol and −5.0 kcal/mol, respectively. These results are consistent with previously published data [7,8].

Altogether, our findings corroborate the proposed antidiabetic effect of *P. granatum* through the inhibition of the major digestive enzymes pancreatic α-amylase and intestinal α-glucosidase and allow us to hypothesize on the phytochemicals responsible for these inhibitions. In agreement with these results, recent papers have reported on the positive action of ellagic acid and *P. granatum* extracts in particular against digestive enzymes [54,55].

### 3.6. Evaluation of the Toxicity of P. granatum Extracts/Fractions in Mice

Oral administration of chloroform, acetone, methanol, and water extracts at a dose of 2000 mg/kg body weight did not produce any toxic effect during the 14 days of the examination. The animals appeared healthy with no clinical symptoms or mortality.

Our results are identical to those published earlier [45,56]. The authors confirmed that there were no toxic effects of the hydroalcoholic extract of the *P. granatum* bark from India and North Jordan at a dose of 2000 mg/kg body weight. Furthermore, the administration of Iranian *P. granatum* bark extract did not cause any toxic signs or death [57]. Another acute toxicity experiment carried out in checks and reported by the authors of [21] revealed that Indian *P. granatum* bark extracts were not lethal at a dosage of 2000 mg/kg body weight.

## 4. Conclusions

Most of the curative effects of the pomegranate fruit have been ascribed to its secondary metabolites. In this study we found that the bioactive compounds obtained from *P. granatum* bark extract by Soxhlet may be beneficial for antidiabetic activity. To investigate this assumption, all extracts were used for phytochemical assays and tested for their abilities to inhibit the activity of both digestive enzymes *α*-glucosidase and *α*-amylase. Chemical compounds were identified by HPLC-DAD. According to our data, the acetone extract showed an underlined inhibitory activity towards the two enzymes which exceeded that of acarbose. Correlation analysis coupled with additional characterization using different approaches (IC_50_ determination, binding, and inhibition modes molecular docking) supported the probable roles of (−)-catechin, ellagic acid, vanillin, and vanillic acid in the inhibitory activity of pomegranate extracts and fractions. The toxicity evaluation of the extracts demonstrated the absence of any oral acute toxicity in mice. Considering the significant gastrointestinal side effects of the standard drug Acarbose, natural inhibitors that might enhance or perhaps replace this drug are now being investigated as possible therapeutic options. Our results support the traditional use of pomegranates to treat DM, propose an inhibitory action on digestive enzymes that could support its hypoglycemic effect, and suggest that pomegranate extracts and/or derived phytochemicals could be promising for the prevention and treatment of T2DM.

## Figures and Tables

**Figure 1 foods-11-01353-f001:**
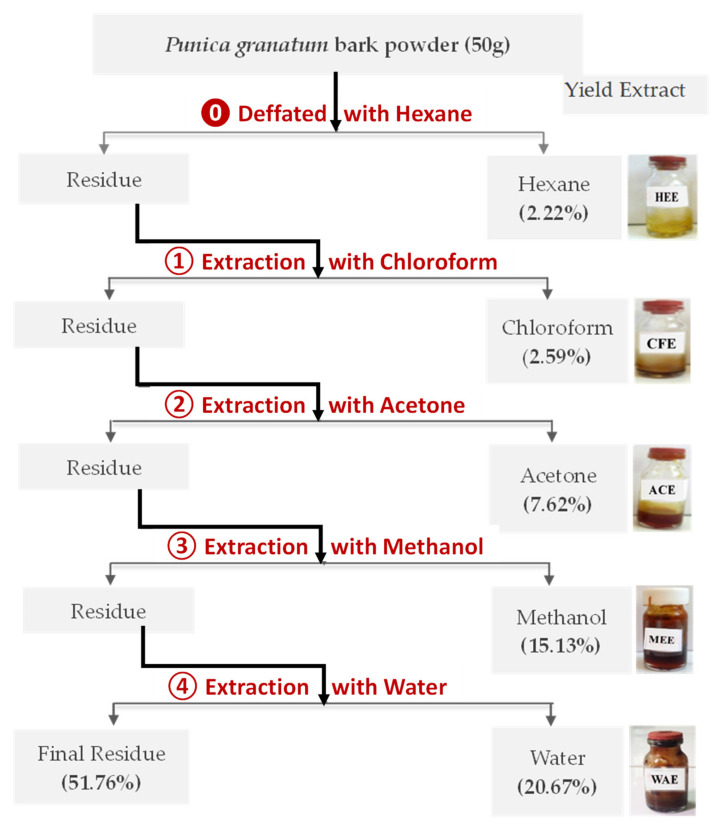
Soxhlet extraction procedure of *Punica granatum* bark and yield of each extract first defatted with hexane (HEE), and then sequentially extracted using solvents with increasing polarities: chloroform (CHE), acetone (ACE), methanol (MEE), and water (WAE).

**Figure 2 foods-11-01353-f002:**
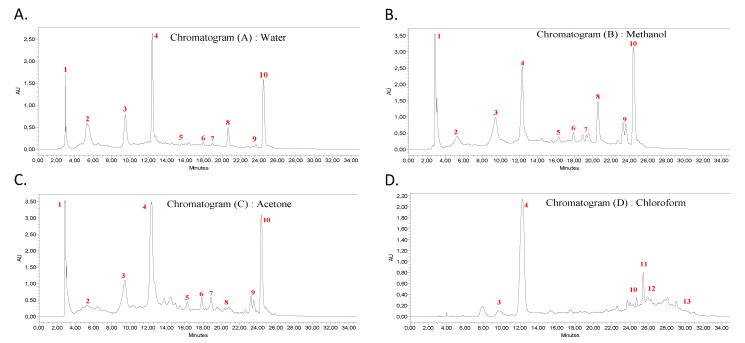
HPLC Chromatograms, visualized at 254 nm, of *P. granatum* bark Extracts: (**A**) Water (WAE), (**B**) Methanol (MEE), (**C**) Acetone(ACE), (**D**) Chloroform (CHE), and (**E**) molecular structures of the main phytochemicals:(1) Punicalin, (2) Gallic acid, (3) Punicalagin a, (4) Punicalagin b, (5) Catechin, (6) Vanillic acid, (7) Caffeic acid, (8) Vanillin, (9) Rutin, (10) Ellagic acid, (11) p-Coumaric acid, (12) Naringenin, (13) Phlorizin.

**Figure 3 foods-11-01353-f003:**
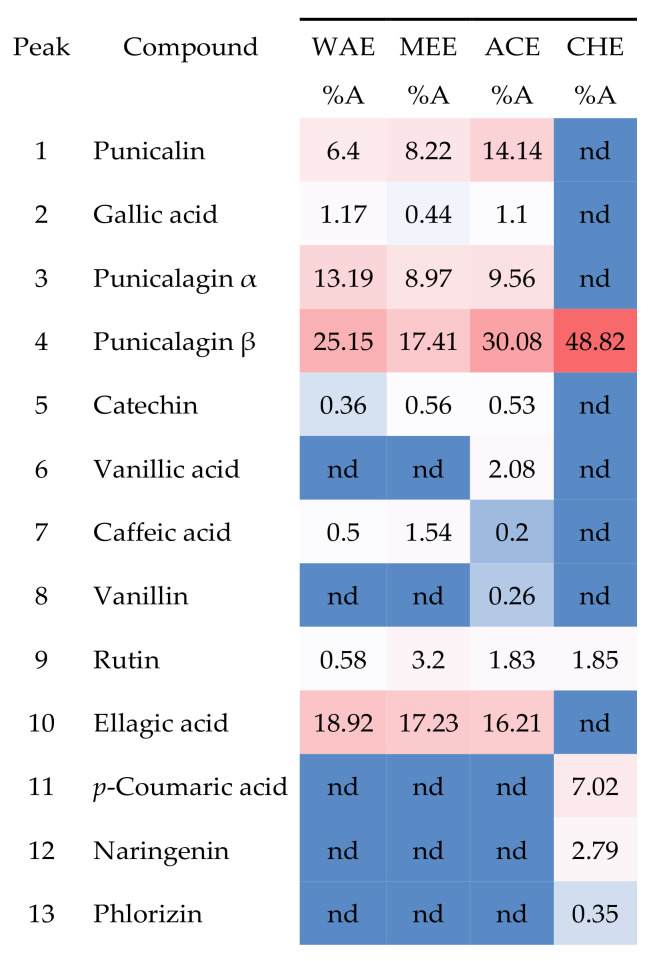
Chemical compounds of *P. granatum* bark extract in all solvents using HPLC-DAD, expressed as a relative percentage of the corresponding peaks areas (%A). Color indicates the relative quantity (red: high, white: medium; blue: low or not detected). nd: not detected; color is referred to the relative quantity of each phytochemical from low (blue), medium (white) to high (red).

**Figure 4 foods-11-01353-f004:**
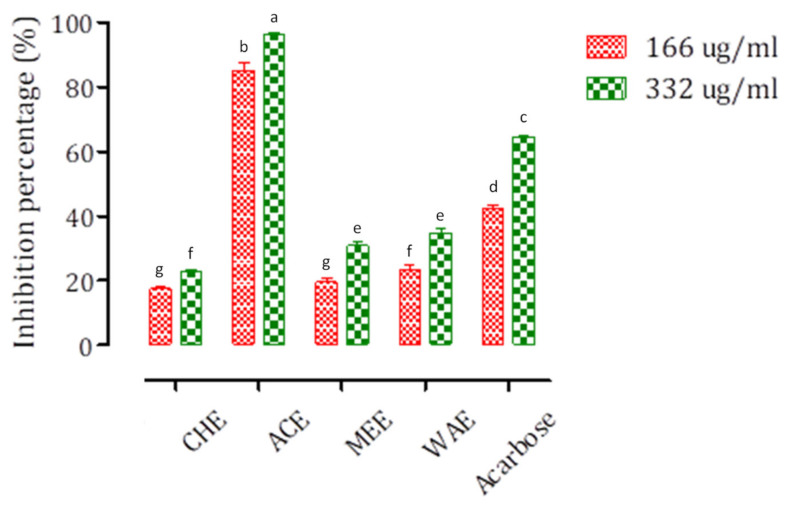
Inhibition of intestinal α-glucosidase by *P. granatum* bark extracts and Acarbose at two concentrations (166 and 332 μg/mL). Different letters indicate significant differences at *p* < 0.05.

**Figure 5 foods-11-01353-f005:**
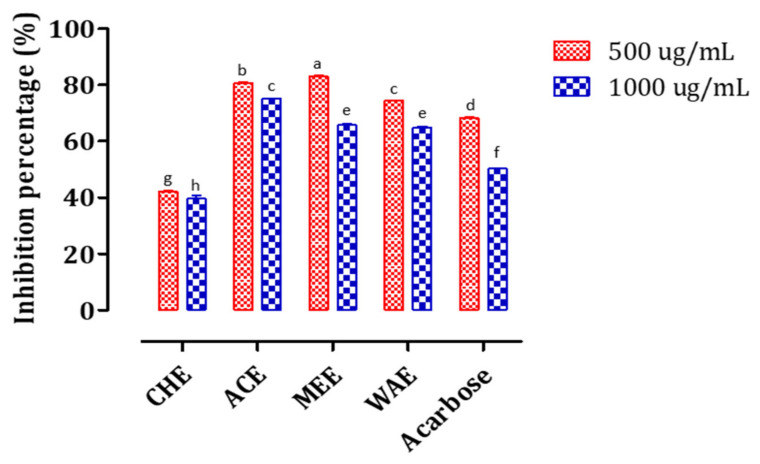
Inhibition of pancreatic α-amylase by *P. granatum* bark extracts and Acarbose at two concentrations (500 and 1000 μg/mL). Different letters indicate significant differences at *p* < 0.05.

**Figure 6 foods-11-01353-f006:**
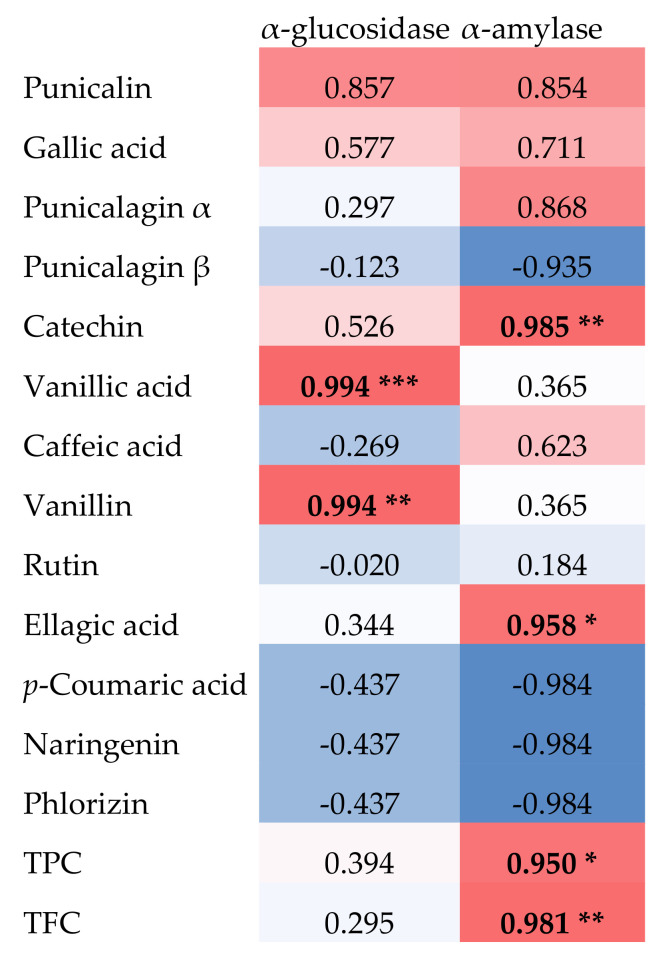
Correlation analysis (Pearson coefficient correlation) linking the phytochemicals and the *α*-glucosidase and α-amylase inhibition capacities of the extracts from *P. granatum* bark. *** significant *p* < 0.001; ** significant *p* < 0.01; * significant *p* < 0.05. Color is referred to the relative quantity of each phytochemical from low (blue), medium (white) to high (red).

**Figure 7 foods-11-01353-f007:**
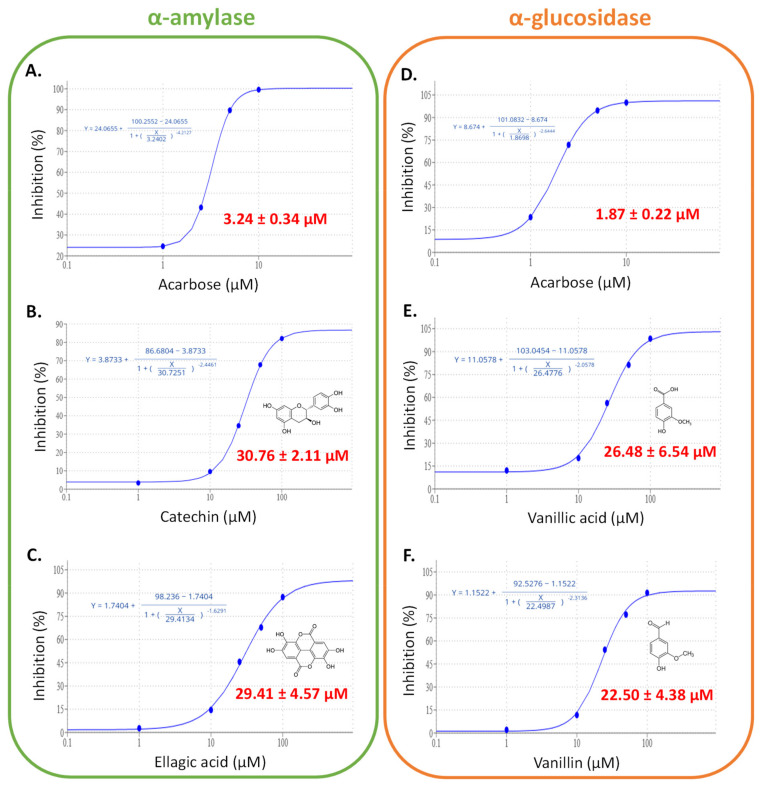
IC_50_ curves (with background correction (subtract smallest response) for the inhibition of intestinal α-amylase by acarbose (**A**), (−)-catechin (**B**) and ellagic acid (**C**), pancreatic α-amylase by acarbose (**D**), vanillic acid (**E**), and vanillin (**F**). Each experiment was performed in triplicate (curves without background correction (subtract smallest response) in order to show error bars are depicted in Appendix A).

**Figure 8 foods-11-01353-f008:**
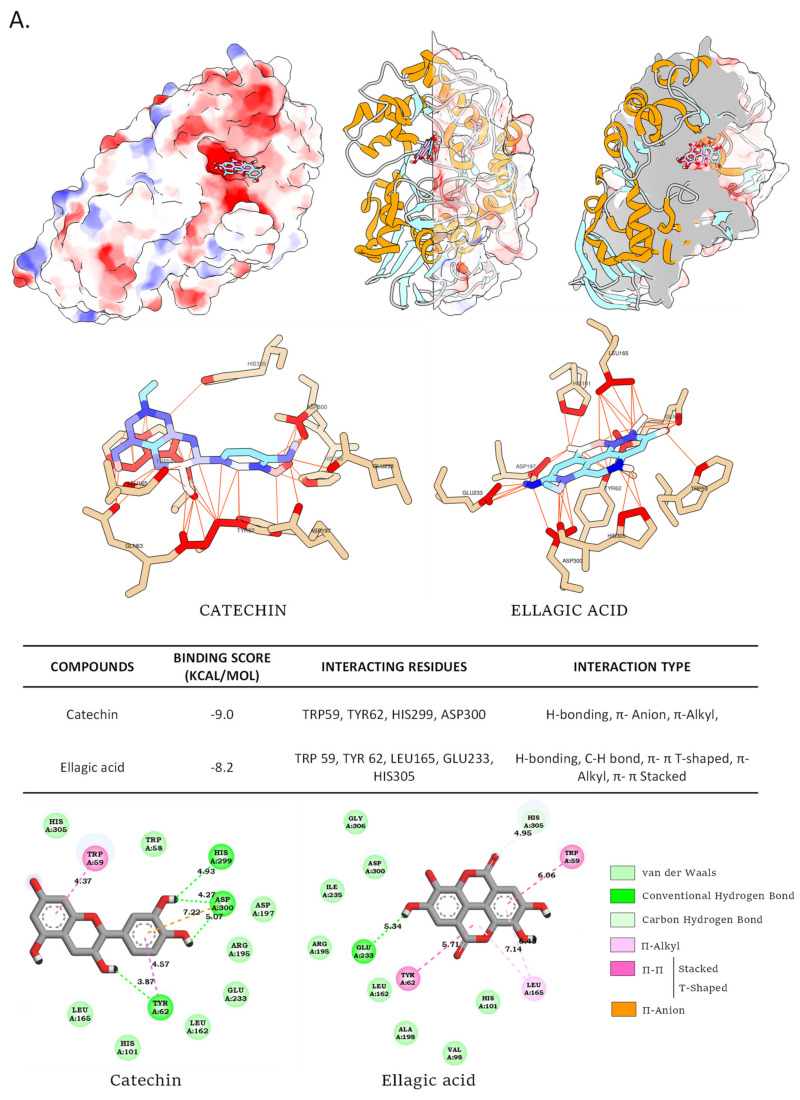
Molecular docking simulations of binding and interaction mode of (**A**). (−)-catechin and ellagic acid to the intestinal α-amylase, and (**B**). vanillic acid and vanillin to the human pancreatic α-amylase.

**Table 1 foods-11-01353-t001:** Qualitative phytochemical analysis of five *P. granatum* bark (PGP) extracts.

	Chloroform (CHE)	Acetone (ACE)	Methanol (MEE)	Water (WAE)
Terpenoids	+	−	−	−
Steroids	+	−	−	−
Alkaloids	−	−	−	+
Flavonoids	−	+	+	+
Phenolics and tannins	−	+	+	+
Coumarins	+	−	−	−
Saponins	−	−	−	−
Anthraquinones	−	−	−	−

“+” and orange color: Present; “−“ and white color: Absent.

**Table 2 foods-11-01353-t002:** Quantitative phytochemical analysis of total phenolic and flavonoid contents in the *P. granatum* bark (PGP) extracts (expressed in mg of GAE and QE /100 g DW).

Extracts	Total Phenolic Compounds (TPC) (mg of GAE/100 g DW)	Total Flavonoid Compounds (TFC) (mg of QE/100 g DW)
Chloroform (CHE)	11.86 ± 1.42 ^d^	7.17 ± 2.064 ^c^
Acetone (ACE)	183.92 ± 3.21 ^b^	117.08 ± 5.32 ^b^
Methanol (MEE)	232.45 ± 9.67 ^a^	147.68 ± 3.03 ^a^
Water (WAE)	131.55 ± 1.19 ^c^	115.89 ± 3.04 ^b^

GAE: gallic acid equivalent; QE: quercetin equivalent; DW: dried weight. Different superscript letters indicate significant differences at *p* < 0.05.

## Data Availability

All the data supporting the findings of this study are included in this article.

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
