# Peer review of "Phytochemical Analysis, α-Glucosidase and α-Amylase Inhibitory Activities and Acute Toxicity Studies of Extracts from Pomegranate (Punica granatum) Bark, a Valuable Agro-Industrial By-Product"

_foods, 2022, doi:10.3390/foods11091353_

Round 1

Reviewer 1 Report

After reviewing the manuscript entitled “Phytochemical Analysis, α-Glucosidase and α-Amylase Inhibitory Activities and Acute Toxicity Studies of Extracts from Pomegranate (Punica granatum) Bark, a Valuable Agro-Industrial By-Product”, some modifications must be considered.

The subject of the manuscript is well chosen, but a thorough revision of the manuscript must be carried out .

I would recommend writing the article in third person. Therefore, make changes to eliminate “we”, “our” or “us” … review the entire manuscript and keep consistency.

P. Granatum: The G of granatum is with a lower case. Review the entire article

Line 196: simple -> sample?

When is it used “0.3% DMSO or distilled water (control)”??

What do the different superscript letters mean?

Figures: Include all the dose-response curves for glucosidase inhibition in the SAME graph, in order to be able to compare them. Also include data from pomegranate extracts, which are the object of study.

The same for amylase inhibition.

Have replications been made? No deviation is seen in the graphs.

Discussion:

A deeper discussion should be carried out, since these properties of the compounds isolated from the extract have already been reviewed. In addition, it should be considered in the speech that these compounds are metabolized into urolithins, which have already shown potential beneficial properties for metabolic syndrome.

  • https://pubmed.ncbi.nlm.nih.gov/29604377/
  • https://publons.com/publon/12972883/

Include these and others references in a deeper discussion.

Author Response

Dear Editors, Dear Reviewers,

Thank you for giving us the opportunity to improve our manuscript by the revised version and thank to your useful comments.

We really appreciate Reviewers’ comments. We hope this revision will satisfy reviewers queries, and that our work will be considered for publication in Foods.

With kind regards

Dr Hano, Dr Mimouni and the co-Authors

REVIEWER 1: After reviewing the manuscript entitled “Phytochemical Analysis, α-Glucosidase and α-Amylase Inhibitory Activities and Acute Toxicity Studies of Extracts from Pomegranate (Punica granatum) Bark, a Valuable Agro-Industrial By-Product”, some modifications must be considered.

The subject of the manuscript is well chosen, but a thorough revision of the manuscript must be carried out.

Authors: Thank you very for your comments. Please find below and in the revised version in Yellow our responses to your comments.

REVIEWER 1: I would recommend writing the article in third person. Therefore, make changes to eliminate “we”, “our” or “us” … review the entire manuscript and keep consistency.

Authors: thank you. We revised it accordingly.

REVIEWER 1: P. Granatum: The G of granatum is with a lower case. Review the entire article

Authors: Thank you, you are right. We revised it accordingly.

REVIEWER 1: Line 196: simple -> sample?

Authors: It’s sample, we revised it accordingly.

REVIEWER 1: When is it used “0.3% DMSO or distilled water (control)”??

Authors: We revised it accordingly. It’s distilled water for the negative control and not DMSO

REVIEWER 1: What do the different superscript letters mean?

Authors: Different superscript letters indicate significant differences at p< 0.05, as indicated in each figure legend.

REVIEWER 1: Figures: Include all the dose-response curves for glucosidase inhibition in the SAME graph, in order to be able to compare them. Also include data from pomegranate extracts, which are the object of study.

Authors: Thank you for your remark. However, Quest Graph™ ED50 Calculator does not allow the inclusion of error bars with the background correction mode (Figure S2). Therefore, to include error bars, we have provided the ED50 curves without background correction on the same graph to allow easier comparison as requested by the Reviewer. Note that ED50 can be calculate only with pure compound, not with an extract with is a complex mixture including potential inhibitors, activators and/or inactive compounds.

The same for amylase inhibition.

Authors: Thank you for your remark. However, Quest Graph™ ED50 Calculator does not allow the inclusion of error bars with the background correction mode (Figure S2). Therefore, to include error bars, we have provided the ED50 curves without background correction on the same graph to allow easier comparison as requested by the Reviewer. Note that ED50 can be calculate only with pure compound, not with an extract with is a complex mixture including potential inhibitors, activators and/or inactive compounds.

Have replications been made? No deviation is seen in the graphs.

Authors: As indicated in Figure 7 legend, “Each experiment was done in triplicate”.

Discussion:

A deeper discussion should be carried out, since these properties of the compounds isolated from the extract have already been reviewed. In addition, it should be considered in the speech that these compounds are metabolized into urolithins, which have already shown potential beneficial properties for metabolic syndrome.

  • https://pubmed.ncbi.nlm.nih.gov/29604377/
  • https://publons.com/publon/12972883/

Include these and others references in a deeper discussion.

Authors: Thank you very much for your advice. These papers have been included for discussion of our results.

Reviewer 2 Report

Pomegranate (Punica granatum) is recognized as a therapeutic fruit possessing several different groups of bioactive compounds. All compartments of pomegranate (P. granatum) like seed, peel, juice, leaves, and bark provide pharmacologic activities.  Thus, this work is interesting and scientifically relevant. In general, the manuscript is well written, properly formatted and with interesting results from the perspective of the development of scientific knowledge. Very current cited bibliographic references. Only a few changes have to be made.

Review: Phytochemical Analysis, α-Glucosidase and α-Amylase Inhibitory Activities and Acute Toxicity Studies of Extracts from 3 Pomegranate (Punica granatum) Bark, a Valuable Agroindustrial By-Product

Considerations:

Pomegranate (Punica granatum) is recognized as a therapeutic fruit possessing several different groups of bioactive compounds. All compartments of pomegranate (P. granatum) like seed, peel, juice, leaves, and bark provide pharmacologic activities.  Thus, this work is interesting and scientifically relevant. In general, the manuscript is well written, properly formatted and with interesting results from the perspective of the development of scientific knowledge. Very current cited bibliographic references. Only a few changes have to be made.

Reviewer suggestions:

Line 69: Lowercase letter replacement (Apple and Grain)

Line 69/70: Punicaceae family - the botanical family name is never written in italics.

Line 84: “chemical families” is not the correct way to write. Please replace with “chemical groups”.

Line 95: Lowercase in all solvent names.

Line 102: After drying, was the moisture control of the samples carried out? Why did you dried only for 48h at 40ºC? There should be an explanation for the conditions chosen.

Line 104: Remove parentheses in the sample mass value.

Line 108: The figure caption is missing. Also, the red letters in Figure 1 are not the same style as the rest of the text. Should be formatted.

Author Response

Dear Editors, Dear Reviewers,

Thank you for giving us the opportunity to improve our manuscript by the revised version and thank to your useful comments.

We really appreciate Reviewers’ comments. We hope this revision will satisfy reviewers queries, and that our work will be considered for publication in Foods.

With kind regards

Dr Hano, Dr Mimouni and the co-Authors

REVIEWER 2: Pomegranate (Punica granatum) is recognized as a therapeutic fruit possessing several different groups of bioactive compounds. All compartments of pomegranate (P. granatum) like seed, peel, juice, leaves, and bark provide pharmacologic activities.  Thus, this work is interesting and scientifically relevant. In general, the manuscript is well written, properly formatted and with interesting results from the perspective of the development of scientific knowledge. Very current cited bibliographic references. Only a few changes have to be made.

Authors: Thank you very for your comments. Please find below and in the revised version in Yellow our responses to your comments.

REVIEWER 2: Line 69- Lowercase letter replacement (Apple and Grain)

Authors: Thank you. We revised it accordingly

REVIEWER 2: Line 69/70- Punicaceae family - the botanical family name is never written in italics.

Authors: Thank you, you are right. We revised it accordingly

REVIEWER 2: Line 84- “chemical families” is not the correct way to write. Please replace with “chemical groups”.

Authors: Thank you. We revised it accordingly

REVIEWER 2: Line 95- Lowercase in all solvent names.

Authors: Thank you. We revised it accordingly

REVIEWER 2: Line 102- After drying, was the moisture control of the samples carried out? Why did you dried only for 48h at 40ºC? There should be an explanation for the conditions chosen.

Authors: This temperature and drying time are necessary and sufficient to preserve and keep the chemical constituents of pomegranate peel powder unchanged during all research work. As a result, the fine powder will be stored away from light at 4°C in borosilated glass bottles for future used.

REVIEWER 2: Line 104- Remove parentheses in the sample mass value.

Authors: Thank you. We revised it accordingly

REVIEWER 2: Line 108- The figure caption is missing. Also, the red letters in Figure 1 are not the same style as the rest of the text. Should be formatted.

Authors: Thank you. We revised it accordingly

Round 2

Reviewer 1 Report

But why are different superscript letters used? and the significant difference is between what?

I must say again that the dose-response curves (a, b and c) should be plotted on the same graph. Same for glucosidase.

The ED50 can of course be calculated for extracts (mg/mL). In addition, the activity of the extract should be plotted together with the reference compounds to observe the differences.

There are many programs that can be used for graphing and calculating IC50.

For the rest, the article is very interesting.

Author Response

Responses to Reviewer:

Reviewer: But why are different superscript letters used? and the significant difference is between what?

Authors: Different superscript letters indicate significant differences at p< 0.05, as indicated in each figure legend.

Reviewer: I must say again that the dose-response curves (a, b and c) should be plotted on the same graph. Same for glucosidase.

Authors: Thank you, but we already provided these dose-response curves (a, b and c) plotted on the same graph for both enzymes in supplementary materials (Figure S2 of the revision 1, now Figure S2 of the present revision).

Reviewer: The ED50 can of course be calculated for extracts (mg/mL). In addition, the activity of the extract should be plotted together with the reference compounds to observe the differences. There are many programs that can be used for graphing and calculating IC50.

Authors: Thank you. We have provided the requested IC50 plots for each extract as supplementary Figure S2 of the present revision.

Reviewer: For the rest, the article is very interesting.

Authors: Thank you very much for your appreciation, we hope the present revision will satisfy you and that the present version will be consider for publication in Foods.